

**Forest liming in the face of climate change: the implications of restorative liming on soil organic**

2 **carbon in mature German forests**

Oliver van Straaten[1], Larissa Kulp[1], Guntars O. Martinson[2], Dan Paul Zederer[1, 3], Ulrike Talkner[1*]

4  1.  Northwest German Forest Research Institute, Grätzelstr. 2, D-37079 Göttingen, Germany

   2.  Soil Science of Tropical and Subtropical Ecosystems, University of Göttingen, D-37077

6      Göttingen, Germany

   3.  Saxon State Office for Environment, Agriculture and Geology, Department of Agriculture,

8      Waldheimer Str. 219, D-01683 Nossen, Germany

   * Correspondence email: ulrike.talkner@nw-fva.de





10 **Abstract**

Forest liming is a management tool that has and continues to be used extensively across northern

12 Europe to counteract acidification processes from anthropogenic sulfur and nitrogen (N) deposition.

In this study, we quantified how liming affects soil organic carbon (SOC) stocks and attempt to

14 disentangle the mechanisms responsible for the often-contrasting processes that regulate net soil

carbon (C) fluxes. Using a paired-plot experimental design we compared SOC stocks in limed plots with

16 adjacent unlimed control plots at 28 experimental sites to 60-cm soil depth in mature broadleaf and

coniferous forests across Germany. Historical soil data from a subset of the paired experiment plots

18 was analyzed to assess how SOC stocks in both control and limed plots had changed between 1990

and 2019.

20 Overall, we found that forest floor C stocks have been accumulating over time, particularly in the

control plots. Liming however largely offsets this organic layer buildup, which means that nutrients

22 remain mobile and are not bound in soil organic matter complexes. Results from the paired plot

analysis showed that forest floor C stocks were significantly lower in limed plots than the control

24 (-34 %, $-8.4 \pm 1.7$ Mg C ha$^{-1}$), but did not significantly affect SOC stocks in the mineral soil, when all sites

are pooled together. In the forest floor layers, SOC stocks exhibited an exponential decrease with

26 increasing pH, highlighting how lime-induced improvements in the biochemical environment stimulate

organic matter (OM) decomposition. Nevertheless, for both forest floor and mineral soils, the

28 magnitude and direction of the belowground C changes hinged directly on the inherent site

characteristics, namely, forest type (conifer versus broadleaf), soil pH, soil texture and the soil SOC

stocks. On the other hand, SOC stock decreases were often offset by other processes that fostered C

accumulation, such as improved forest productivity or increased carbon stabilization, which

correspondingly translated to an overall variable response by SOC stocks, particularly in the mineral

soil.

Lastly, we measured soil carbon dioxide ($CO_2$) and soil methane ($CH_4$) flux immediately after a re-liming

event at three of the experimental sites. Here, we found that (1) liming doubles $CH_4$ uptake in the long-

36 term, (2) highlighted that soil organic matter mineralization processes respond quickly to liming,

though the duration and size of the $CO_2$ flush varied between sites, and (3) lime-derived $CO_2$

contributed very little to total $CO_2$ emissions over the measurement period.



## 1. Introduction

Millions of hectares of forest have been limed in Germany and across northern Europe over the last few decades to counteract soil acidificatin processes derived from anthropogenic sulfur (S) and nitrogen (N) deposition. Soil acidification is responsible for hindering organic matter decomposition processes and concomitantly immobilizing nutrients and carbon (Shen et al., 2021).The application of lime on acidic soils, as either calcium carbonate ($CaCO_3$) or dolomite ($CaMg(CO_3)_2$) elicits a strong biochemical response by lowering soil acidity, reducing both aluminum (Al) and manganese toxicity and increasing the soil's buffering capacity. These changes subsequently drive a cascade of ecosystem responses, with implications on soil fertility, forest productivity, stand vitality and litter decomposition (Derome et al., 2000, Kreutzer, 1995), which in turn correspondingly affect the ecosystem carbon (C) balance (Melvin et al., 2013; Persson et al., 2021) and soil greenhouse gas (GHG) budgets. The direction and magnitude of ecosystem responses to liming depends on numerous factors, including: (1) the inherent soil characteristics of the site (soil acidity, soil texture, the chemical make-up of the forest floor layer), (2) vegetation characteristics (species distributions, tree density, and stand age), the (3) lime application (type and quantity of lime, and frequency of liming) and (4) ongoing acidification from recent N and S deposition. In this context, both above- and below-ground carbon stocks have been shown to have quite variable responses to liming (Court et al., 2018; Lundström et al., 2003; Melvin et al., 2013; Persson and Ahlström, 1990; Persson et al., 2021).

While, it is broadly reported  that liming stimulates soil microbial activity leading to accelerated soil organic matter (SOM) decomposition (Andersson and Nilsson, 2001, Kreutzer, 1995), some studies report either no change in litter and forest floor decomposition (Smolander et al., 1996) or even forest floor accumulations (Derome et al., 2000; Melvin et al., 2013). Soil organic carbon (SOC) stock gains as a result of liming can be attributed to different drivers. First, earthworm abundance is known to increase after liming (Persson et al., 2021) which, by actively incorporating and binding SOM with the mineral soil improves physical properties, such as soil structure and aggregate stability (Bronick and Lal, 2005). Second, physicochemical properties are also affected. Liming induced changes in nutrient-stoichiometry may enhance cation mediated cross-linking between SOM compounds and divalent calcium (Ca) or magnesium (Mg) ions (Kalbitz et al., 2000). It has also been shown that higher soil Ca availability has been shown to increase lignin contents in leaf litter which makes litter more recalcitrant and resistant to decomposition (Eklund and Eliasson, 1990; Xing et al., 2021). Third, liming may introduce nutrient imbalances (such as phosphorus) on decomposer communities and trees (Melvin et al., 2013) that may decrease microbial breakdown of SOM. In addition, liming may induce shifts in microbial community structures, and decrease microbial abundance (Melvin et al., 2013). Lastly, liming-induced improvements in nutrient availability (Jansone et al., 2020; Long et al., 2015), may



74 increase ecosystem productivity which correspondingly can increase SOM inputs from aboveground (e.g. leaf litter (Lin et al., 2015)) and belowground sources (e.g. root detritus).

76 In this study, we quantified the magnitude of SOC stock changes resulting from forest liming activities, with the explicit intent to better understand the implications of liming on forest soil greenhouse gas

78 (GHG) budgets. Given the lack of a consistent direction in which SOC stocks respond to liming as reported in literature, we attempted to disentangle the mechanisms responsible for the often-

80 contrasting processes that regulate net carbon fluxes in the soil. Lastly, we assessed liming effects across different time scales, ranging from the immediate effects liming has on soil carbon dioxide ($CO_2$)

82 production, to methane ($CH_4$) uptake, to long-term changes in soil carbon stocks measured several decades after liming. The study was implemented at experimental sites in managed mature forests

84 across Germany using both space-for-time substitution and chronosequence approaches.

We hypothesized that liming-induced changes in SOC stocks will be most pronounced at the soil

86 surface. More specifically, we expect that there will be significant decreases in the forest floor layer C stock because SOM decomposition will be stimulated by reduced pH levels. However, these C losses

88 will be offset if not exceeded, by significant gains in SOC stocks in the topsoil because of improved ecosystem productivity, increased fine root biomass in the upper mineral soil horizons and increased

90 earthworm activity, which will improve soil structure thereby protecting SOM from mineralization.

## 2. Methods and Materials

92 ### 2.1 Experimental study sites

Liming effects on soil organic carbon stocks were determined at 28 liming experiment sites distributed

94 across Germany (Figure A1). All sites consisted of mature forest stands whereby all, except one (HLI 2680) were managed, meaning sites were occasionally selectively harvested. Lime was applied in

96 different forms (dolomite ($CaMg(CO_3)_2$) and calcium carbonate ($CaCO_3$)) and in differing quantities, ranging from a total between 2-9 tons per hectare spread over multiple application dates (Table 1).

98 The last lime application at most sites was typically 20 to 30 years prior to our sampling, and therefore findings reflect the long-term effects liming has on belowground carbon. The experiment was

100 conducted using a paired plot design, where each site consisted of a limed plot adjacent to a control plot which was not limed. In total, for this analysis, we sampled nine sites with European beech (*Fagus*

102 *sylvatica* L.), two with common oak (*Quercus robur* L.), 16 with Norway spruce (*Picea abies* L. karst.) and one European red pine (*Pinus sylvestris* L.) site. General site characteristics are described in Table

104 1. At two spruce sites (GOH 155, SEG 244) we only had data from the forest floor layers, and not the mineral soil as soil bulk density data were unavailable. Nitrogen deposition was ascertained from the

106 German Environment Agency (UBA, 2019).



**Table 1:** Site characteristics and liming details of the 28 experimental sites. Soil texture measurements were only made at 16 sites.

| Site name | Experiment and liming details | | | | | Climate | | | | Soil | | |
|---|---|---|---|---|---|---|---|---|---|---|---|---|
| | Plot size (limed/ control) [ha] | Number of times limed | Type of lime † | Lime quantity [Mg ha⁻¹] | ANC‡ [kmol$_c$ ha⁻¹] | Mean annual precip. [mm a⁻¹] | Mean annual temp. [°C] | Elevation [m.asl] | Nitrogen deposition [kg N ha⁻¹ yr⁻¹] | Soil pH (H₂O; 0-5 cm; limed/ control) | Soil base saturation (0-5cm; limed/ control) [%] | Soil texture (30-60 cm; sand / silt / clay) [%] |
| **Beech sites:** | | | | | | | | | | | | |
| Beerfelden 767A | 0.25 / 0.1 | 2 | B, B | 1, 1 | 42 | 977 | 8.9 | 447 | 14 | 4.1 / 3.9 | 30 / 7 | 70 / 18 / 12 |
| Dassel 4227 | 0.2 / 0.1435 | 2 | A, B | 5, 3 | 140 | 1221 | 7.7 | 430 | 19 | 4.8 / 3.7 | 30 / 7 | 50 / 36 / 14 |
| Eutin 402 | 0.25 / 0.25 | 2 | B, B | 3, 3 | 109 | 746 | 8.2 | 55 | 22 | 4.9 / 4.1 | 67 / 11 | n.a. |
| Göhrde 157 | 0.25 / 0.25 | 2 | A, B | 5, 3 | 140 | 733 | 8.8 | 100 | 16 | 4.6 / 3.7 | 40 / 11 | 93 / 4 / 3 |
| Grünenplan 142 | 0.25 / 0.25 | 3 | B(G), B, B | 5, 3, 3 | 133 | 920 | 8.9 | 260 | 19 | 5.2 / 4.1 | 52 / 15 | 4 / 73 / 23 |
| Grünenplan 51 | 0.3 / 0.3 | 1 | B(G) | 5 | 75 | 920 | 8.9 | 330 | 19 | 5.7 / 5.0 | 93 / 73 | n.a. |
| Hess. Lichtenau 2680 | 0.3 / 0.3 | 2 | B, B | 1, 1 | 41 | 970 | 7.3 | 487 | 17 | 4.2 / 4.1 | 10 / 7 | 32 / 50 / 18 |
| Jossgrund 2268 | 0.3 / 0.3 | 2 | B, B | 1, 1 | 41 | 1050 | 8.5 | 385 | 13 | 4.7 / 4.3 | 49 / 16 | 58 / 30 / 12 |
| Sellhorn 34 | 0.15 / 0.15 | 2 | A, B | 5, 3 | 148 | 849 | 8.9 | 110 | 19 | 4.6 / 4.0 | 56 / 13 | 85 / 11 / 4 |
| **Oak sites:** | | | | | | | | | | | | |
| Göhrde 140 | 0.25 / 0.25 | 2 | A, B | 5, 3 | 140 | 733 | 8.8 | 95 | 16 | 4.7 / 4.1 | 37 / 7 | n.a. |
| Sellhorn 66 | 0.4 / 0.4 | 2 | A, B | 5, 3 | 140 | 849 | 8.9 | 110 | 20 | 4.4 / 4.2 | 48 / 17 | n.a. |
| **Spruce sites:** | | | | | | | | | | | | |
| Bad Waldsee | 4.28 / 5.21 | 3 | A, B | 2, 6 | 171 | 970 | 8.6 | 571 | 19 | 5.8 / 3.9 | 91 / 14 | 59 / 28 / 13 |
| Beerfelden 767B | 0.15 / 0.15 | 2 | B, B | 1, 1 | 42 | 977 | 8.9 | 442 | 15 | 3.5 / 3.5 | 8 / 5 | n.a. |
| Dassel 325 | 0.2 / 0.1 | 3 | A, B, B | 5, 3, 3 | 140 | 1221 | 6.9 | 390 | 20 | 4.4 / 3.8 | 46 / 5 | n.a. |
| Ellwangen | 10.24 / 1.32 | 3 | A, B | 3, 6 | 171 | 847 | 8.8 | 482 | 16 | 6.3 / 4.0 | 92 / 24 | 64 / 26 / 10 |
| Freudenstadt | 7.71 / 3.46 | 3 | A, B | 3, 6 | 171 | 1516 | 7.4 | 748 | 13 | 4.6 / 3.7 | 70 / 6 | 75 / 16 / 8 |
| Göhrde 155 * | 0.25 / 0.25 | 2 | A, B | 5, 3 | 140 | 733 | 8.8 | 80 | 18 | - | - | - |
| Heidelberg | 2.13 / 0.82 | 3 | A, B | 3, 6 | 171 | 1193 | 8.8 | 477 | 14 | 6.6 / 3.6 | 98 / 14 | 69 / 22 / 9 |
| Herzogenweiler | 8.28 / 3.28 | 3 | A,, B | 3, 6 | 171 | 1203 | 6.7 | 909 | 12 | 5.9 / 3.8 | 95 / 5 | 55 / 24 / 21 |
| Horb | 8.35 / 2.27 | 3 | A, B | 3, 6 | 171 | 969 | 8.2 | 623 | 12 | 4.7/ 4.1 | 55 / 32 | 44 / 36 / 20 |
| Hospital | 2.59 / 0.51 | 3 | A, B | 3, 6 | 171 | 827 | 8 | 645 | 18 | 5.7 / 3.8 | 91 / 11 | 36 / 45 / 19 |
| Lauterberg 2023 | 0.25 / 0.25 | 3 | D, B, B | 1, 3, 3 | 128 | 1220 | 6.1 | 570 | 22 | 4.9 / 4.1 | 46 / 8 | n.a. |
| Lauterberg 75 | 0.25 / 0.25 | 2 | D, E | 1, 3 | 131 | 1454 | 5.1 | 790 | 25 | 4.4 / 4.3 | 10 / 5 | n.a. |
| Rantzau 50 | 0.2217 / 0.25 | 3 | C, B, B | 3, 3, 3 | 140 | 807 | 8.4 | 35 | 26 | 3.9 / 3.6 | 30 / 8 | n.a. |
| Segeberg 244 * | 0.25 / 0.25 | 3 | B, B, B | 3, 3, 3 | 109 | 800 | 8.3 | 34 | 26 | - | - | - |
| Segeberg 517 | 0.25 / 0.25 | 3 | B, B, B | 3, 3, 3 | 109 | 844 | 8.3 | 20 | 26 | 4.1 / 3.7 | 34 / 11 | n.a. |
| Weithard | 1.25 / 0.59 | 3 | A(F), B | 3, 6 | 171 | 832 | 8.1 | 627 | 16 | 5.1 / 3.8 | 80 / 10 | 35 / 47 / 18 |
| **Pine site:** | | | | | | | | | | | | |
| Göhrde 129 | 0.25 / 0.25 | 3 | A, B, B | 5, 3, 3 | 140 | 733 | 8.8 | 70 | 18 | 4.8 / 4.1 | 49 / 12 | n.a. |

* Only forest floor layer sampled in this plot; † Types of lime: A = Calcium carbonate, B = Dolomite; C = Marl lime, D = Thomas-phosphate, E= Slag lime, F = Potassium sulfate, G = Rock phosphate; ‡ Acid neutralizing capacity



### 2.2 Soil organic carbon stocks

We collected soil and forest floor samples from both limed and control plots from each site at four locations distributed around the plot. Samples were taken from the forest floor (L/O$_f$ and O$_h$) as well

as from the mineral soil at predefined depths (0-5, 5-10, 10-30 and 30-60 cm). Samples of the forest floor and the topsoil (0-30 cm) were taken using a root auger (diameter 8 cm) and samples of the

subsoil (30-60 cm) using a gouge auger (diameter 3 cm). At each of the four sampling locations per plot, three samples were taken for each depth and pooled. Forest floor samples were subsequently

oven dried at 60 °C, sieved (2 mm) and ground, mineral soil samples were oven dried at 40 °C, sieved (2 mm) and ground. Both forest floor and mineral soil samples were then analyzed for carbon (C) and

nitrogen (N) contents using a CN analyzer (Euro EA - CN Elemental Analyzer, HEKAtech GmbH, Wegberg, Germany). Sieved forest floor and mineral soil samples were also analyzed for pH in a 1:2.5

H$_2$O solution and mineral soil samples for exchangeable cations (Ca, Mg, K, Na, Al, Fe, Mn) using an ICP-AES instrument (Thermo Scientific iCAP 7400 Radial, Thermo Scientific, Dreieich, Germany). Base

saturation was calculated as percentage exchangeable base cations of the effective cation exchange capacity (ECEC). Soil texture was determined using the pipette method at 16 experiment sites.

Soil bulk density and the mineral soil dry mass per unit area was determined using a modified version of the Blake and Hartge (1986) core method. Samples were taken at four soil pits per plot for the same

respective depths where samples were taken for chemical analysis. Depending on the size and relative abundance of stones observed in the soil profile, different approaches were employed to estimate the

bulk density of the soil fine-fraction. Methods and equations are described by König et al. (2014). All samples were oven dried at 105 °C for 48 hours and subsequently weighed. Volumes of coarse

fragments were determined using the volume displacement method. For the mineral soil, we calculated the fine earth soil mass per unit area for each respective sampling layer as follows:

Fine earth mass per unit area = BD * (1-stone content) * d * 10         (1)

Where, fine earth soil mass per unit area is in kg m$^{-2}$, BD is the soil bulk density in g cm$^{-3}$, stone content

is relative volumetric coarse fragment content, d is the thickness (depth) of the sampling horizon in centimeters and 10 is a conversion factor for converting g cm$^{-2}$ to kg m$^{-2}$.

The organic layer dry mass per unit area was determined at the same four sampling locations where the samples for the chemical analysis were collected using a root auger (diameter 8 cm). The organic

material from within the auger was collected and separated into the two forest floor layers (L/O$_f$ and O$_h$). Roots and plant debris larger than 2 cm in size were removed from the sample, whereupon

samples were oven dried and weighed (König et al., 2014):

Organic layer dry mass per unit area = (MH *100) / SA *10         (2)



whereby, organic layer mass per unit area is in kg m$^{-2}$, MH is the dry mass of the organic layer in grams, and SA is the surface area that was sampled in cm$^2$, and 10 is a conversion factor for converting to kg

m$^{-2}$. Mineral and forest floor organic carbon stocks were calculated as follows:

$$\text{SOC stock } = \frac{OC * MuA}{100}$$     (3)

whereby, SOC stock is in Mg C ha$^{-1}$, OC is the organic C content in g kg$^{-1}$, MuA is the mass per unit area in kg m$^{-2}$, and 100 is a conversion factor for converting to Mg C ha$^{-1}$.

SOC stocks in the limed plots were corrected for fixed-depth differences incurred because of liming-induced changes in soil bulk density (Figure A2) by using the equivalent soil mass (ESM) approach

described by Wendt and Hauser (2013). This approach fits a cubic spline curve of cumulative organic carbon stocks with the corresponding soil mass of the reference profile.

Effects of liming were evaluated using two approaches. First, the difference in soil C stocks between limed and control plots were calculated to assess the relative differences. Second, a chronosequence

approach was used to assess temporal changes in soil C stocks using historic data, between 1990 and 2019, collected at a subset of the paired experiment sites (forest floor: n = 17, mineral soil: n = 13).

Table A1 (in the Supplement) shows the years when forest floor and mineral soil samples were collected. The change in SOC stocks over time was estimated by calculating the slope of a linear best

fit function of the SOC stock values over time. In this analysis, we assumed that soil density did not change during this time and accordingly we used bulk density estimates from the most recent

measurement date.

### 2.3 Short term effects of liming on soil CO$_2$ and CH$_4$ fluxes

Soil carbon dioxide (CO$_2$) and methane (CH$_4$) fluxes were measured at three beech forest sites (Dassel 4227 (DAS 4227), Sellhorn 34 (SEL 34), Göhrde 157 (GOH 157) to assess both short and long-term

effects of liming. All three sites were freshly re-limed with an equivalent of 3 Mg CaCO$_3$ ha$^{-1}$ in August-September 2020. Accordingly, the measurements made after these liming events reflect the short-

term effects of liming on soil respiration and soil methane fluxes. The soil trace gas fluxes were measured using the vented static chamber method. Round chamber bases (polyvinyl chloride, covering

a ground area of 0.07 m$^2$) were inserted 1–2 cm into the soil surface at four randomly locations within each plot. These chamber bases were covered with polyethylene lid (~22 L headspace volume), from

which gas samples were collected at 20-min intervals for one hour (2, 22, 42 and 62 min) and stored in pre-evacuated 12 mL Labco Exetainers® (Labco Limited, Lampeter, UK). To minimize effects from

diurnal fluctuations we randomized the order the plots were measured during each measurement campaign. Gas samples were analyzed using a gas chromatograph (GC, SRI 8610c, SRI Instruments,



Torrance, USA), equipped with a flame ionization detector to measure $CH_4$ and $CO_2$. The latter gas species was analyzed by converting it to $CH_4$, using a built-in methanizer in the GC. The GC was

calibrated prior to each analysis using three calibration gases (Deuste Steininger GmbH, Mühlhausen, Germany), spanning the concentration range of the field samples. Soil gas fluxes were calculated using

the ideal gas law, based on the linear increase of gas concentrations in the chamber over time and corrected with air temperature and atmospheric pressure measured at the time of sampling.  A

positive flux indicates a net emission, while a negative flux indicates a net consumption. In parallel to the greenhouse gas flux measurements, we also measured air pressure, soil and air temperature and

chamber volume during each measurement.

In early September 2020, we measured soil $CO_2$ and $CH_4$ fluxes at one site (DAS 4227) three times in

the week prior to lime application (on Sep 7, 2020) so as to evaluate baseline fluxes and to determine whether there were long-term effects of previous liming events still evident. After liming, we measured

GHG fluxes weekly for two months (to Nov. 3, 2020) to evaluate immediate effects of the liming. Subsequently, in the spring of 2021, we resumed gas flux measurements on a bi-weekly basis at the

DAS 4227 site, and additionally also commenced measurements at the two other sites (SEL 34, GOH 157). These measurements were made from Mar. 11, 2021 to Jul. 8. 2021.

**2.4 Calculation of lime-derived $CO_2$ emissions**

The proportion of lime-derived $CO_2$ to the overall $CO_2$ flux, was determined using $\delta^{13}C$ stable isotope

approaches and a two-pool mixing model. The $^{13}C$ signature of newly formed $CO_2$ ($\delta n$) between time point t = 1 ($\delta_1$) and t = 2 ($\delta_2$), and the newly formed $CO_2$ fraction at t = 2 is given by the following mass

balance equation (Martinson et al., 2018):

$$\delta_2 = f_n\delta_n + (1 - f_n)\delta_1 \tag{4}$$

The fraction of lime- derived $CO_2$ to total $CO_2$ emissions is calculated following the two-pool mixing model under the assumption that (1) biologically-derived $^{13}CO_2$ is equal between limed and unlimed

plots and (2) $CO_2$ from lime carbonates and from lime-induced respiration is in isotopic equilibrium:

$$f = \frac{(\delta + \delta_0)}{(\delta_1 + \delta_0)} \tag{5}$$

whereby, $\delta$ is the isotopic signature of $^{13}CO_2$ from limed plots, $\delta_0$ the isotopic signatures of $^{13}CO_2$ from unlimed plots, $\delta_1$ the isotopic signature of lime.

Carbon isotope signatures ($\delta^{13}C$) were determined by isotope ratio mass spectrometry at the Centre for Stable Isotope Research and Analysis (KOSI) at the University of Göttingen, Germany.



### 2.5 Statistical analysis

Liming effects on SOC stocks at each soil depth were tested using linear mixed effects (LME) models (Crawley, 2013). In these models, the C stock was the response variable, the treatment (control, limed) was the fixed effect, and the site was the random effect. For the soil trace gas flux measurements, the treatment was considered a fixed effect and the measurement date were considered random effects. Significance levels were tested separately for each site. Differences were considered significant if $P \leq 0.05$ and marginally significant if $P \leq 0.1$. The input C stock data as well as the output model residuals were tested for normality using Shapiro–Wilk test. To gain an insight into the underlying factors regulating C stocks in the control (unamended plots) and the relative changes in C stocks as a result of liming, we used Spearman's rank correlation analyses to assess how C stocks correlated with climatic parameters, stand parameter as well as the inherent soil properties (of the control plot) and the liming induced changes in soil properties (difference between limed and control plots). The goodness of fit of the non-linear best-fit models were assessed using Pearson correlation analyses between model-predicted values and measured values. All statistical analyses were carried out using R, version 4.0.02 (R Core Team, 2020).

### 3. Results

#### 3.1 SOC stocks in the control plots: magnitude and drivers

There was a large variability in SOC stocks across the experimental sites, ranging between 49 and 366 Mg C ha$^{-1}$ (forest floor to 60 cm, in the control plots). In the soil profile, SOC content was highest in the forest floor layer and decreased with soil depth (Figure A3a). In the control plots, 23 % of the total SOC stock was found in the forest floor layer, 27 % in the topsoil (0-10 cm), and the remaining 50 % was found below 10 cm depth (10-60 cm) (Figure A3b). Coniferous forests stored approximately 38 % more carbon than broadleaf forests (conifer: 157 ± 17 Mg C ha$^{-1}$ (mean ± standard error (SE)), broadleaf: 97 ± 9 Mg C ha$^{-1}$), where differences were most pronounced in the forest floor L/O$_f$ horizon and below 10 cm soil depth. Soil bulk density was lowest at the soil surface (0-5 cm) and increased with soil depth (Figure A2a). SOC stocks in the mineral soil correlated significantly with soil chemical and physical properties, but not with climatic variables such as temperature, precipitation, or elevation (Table A2). In the forest floor layers, SOC stocks were correlated with both N deposition and pH. For the latter, there was an exponential decrease in the SOC stocks with increasing pH (Figure 1), where, particularly in the L/O$_f$ layer, there was large decline in SOC stocks when pH increased from 3.5 to 4.5. Next, N-deposition exhibited a significant positive correlation with SOC stock in the L/O$_f$ horizon, whereby





these effects were only evident in coniferous forests (Figure A4a). This trend was largely driven by the
strong linear correlation present between C content and N deposition (Figure A4b), and although the
mass of the L/O$_f$ horizon correspondingly increased with N deposition, the most increases were only
consistent when N deposition was higher than 25 kg N ha$^{-1}$ yr$^{-1}$ (n = 4) (Figure A4c).

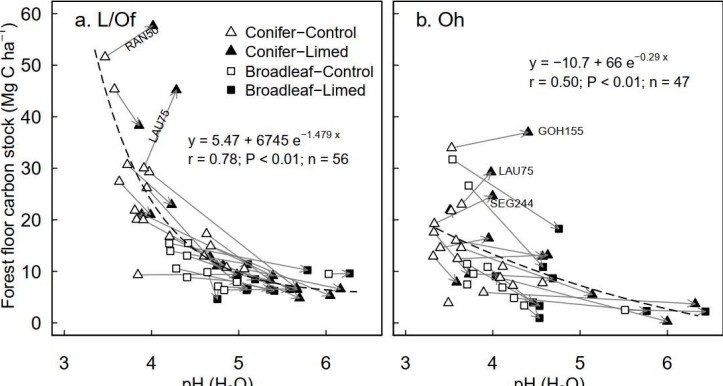

**Figure 1:** Relationship between the pH of the forest floor layer and the carbon stock of the (a) L/O$_f$ and
(b) O$_h$ horizons. The arrows indicate the change from control to limed plots. Missing arrows (in b.) are
because the O$_h$ horizon disappeared at those sites as a result of liming. The r is the Pearson correlation
coefficients between observed and fitted values. RAN50 is Rantzau 50, LAU75 is Lauterberg 75,
GOH155 is the Göhrde 155, SEG 244 is the Segeberg 244 site.

In the mineral soil, SOC stocks correlated with soil texture fractions. This was evident in the significant
negative correlations between SOC stock and sand contents at 0-5 cm and 5-10 cm, as well as the
positive correlation with clay content at 10-30 cm (Table A2). In the subsoil (30-60 cm), SOC stocks
exhibited a similar exponential decay relationship with soil pH as the forest floor layers (data not
shown).

**3.2 SOC stock response to liming: chronosequence approach**

At a subset of experimental sites where historical data were available, forest floor C stocks in the
control plots increased in time (0.5 ± 0.1 Mg C ha$^{-1}$ yr$^{-1}$; Figure 2a), whereby the increases were largely
driven by C accumulations at coniferous forest site (0.8 ± 0.3 Mg C ha$^{-1}$ yr$^{-1}$). Although forest floor SOC
stocks in the limed plots also increased over time, the accumulation rates in the L/O$_f$ horizon were
significantly lower than the control (Figure 2b). In the mineral soil, there were no significant changes
in SOC stocks at any depth during this period. Nevertheless, when C stock change rates were compared
between limed and control plots, liming did bolster C accumulation rates slightly at 5-10 cm depth.



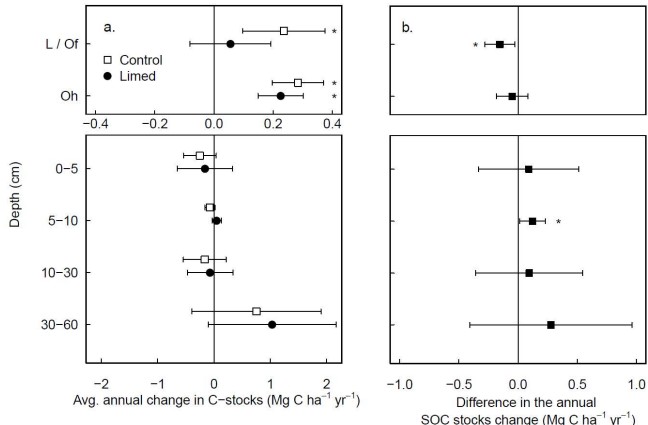

**Figure 2:** Average (± 95 % T-test-confidence interval) annual changes in SOC stocks experienced over
the last two decades in a) both the control and limed plots and b) the difference between the limed
and control plots. Statistical significance was tested using LME models for each respective soil depth /
layer at P ≤ 0.05 (*).

### 3.3 SOC stock response to liming: paired plot approach

Total SOC stocks (forest floor to 60 cm) were comparable between the limed (126 ± 12 Mg C ha$^{-1}$) and

control plots (132 ± 12 Mg C ha$^{-1}$) (Figure A3b). In the forest floor layer, liming significantly reduced

SOC stocks by 34 ± 12 % (equivalent to 8.4 ± 3.6 Mg C ha$^{-1}$, Figure 3a), which reflects reductions in both

C content (-8.8 ± 2.4 %) as well as the forest floor dry mass (-26.1 ± 6.7 %). Both broadleaf and

coniferous forests had similar SOC losses in the forest floor layer both in magnitude (broadleaf: -

8.1 ± 4.8 Mg C ha$^{-1}$, coniferous: -8.6 ± 2.7 Mg C ha$^{-1}$) and in overall proportion of the forest floor SOC

stock (Figure 3b).

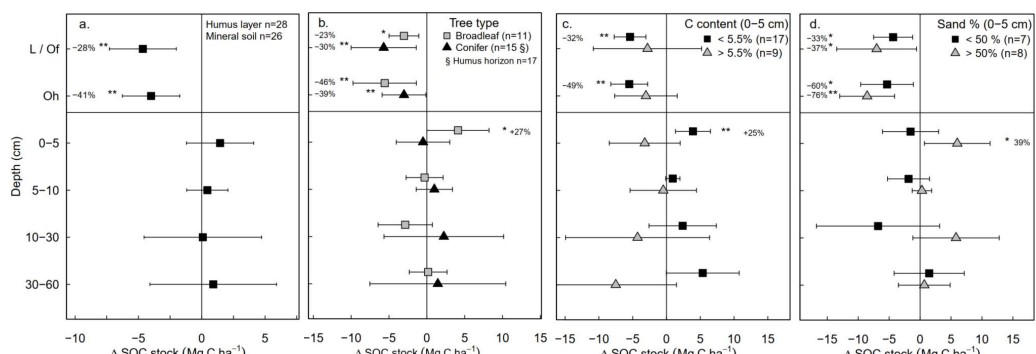

**Figure 3:** Changes in mean soil organic carbon stock (limed - control) as a result of liming (a) for all
plots in the experiment, and classified by (b) tree types and (c) inherent C content of the control plots
and (d) site sand percent from 0-5 cm depth. Error bars indicate the 95 % confidence intervals based
on Student's T distribution. Statistical significance was tested using LME models for each respective
soil depth / layer and grouping at P ≤ 0.05 (*), and P ≤ 0.01 (**).



The liming quantities which are responsible for the changes in soil pH, exhibited a negative linear relationship with SOC stock changes (Figure A5), indicating that higher liming dosages result in larger SOC losses. In the forest floor layer, the proportion of C losses or C gains (at a few sites) could further be explained by the initial C stock present on the site (control plot C stock), whereby the C losses were largest at sites with medium amounts of stored C (between 20 and 35 Mg C ha$^{-1}$), and less pronounced (or even positive) at sites with either little or large C stocks present in the reference state (Figure 4e).

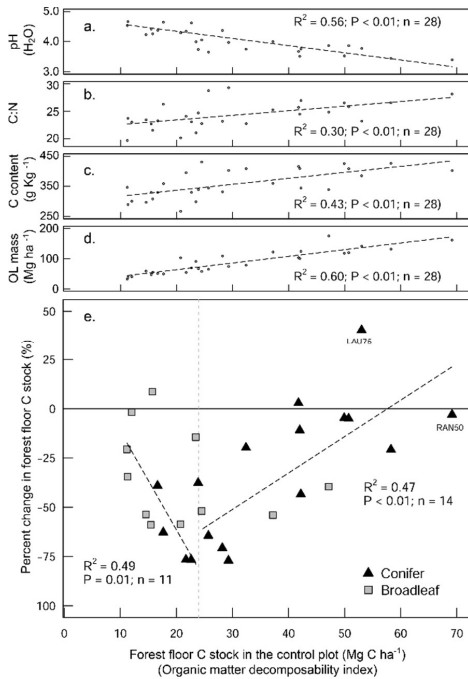

**Figure 4:** Scatterplot diagrams showing how the forest floor SOC stocks in the control plots relate to a) pH (H$_2$O), b) the C:N ratio, c) the C content of the forest floor layer and d) the forest floor (organic layer (OL)) biomass. These four parameters show how the forest floor C stock in the control plots is a good index of organic matter decomposability. The scatterplot in e) shows the percentage change in the forest floor C stock as a result of liming in relation to the forest floor decomposability index (forest floor C stocks of the control plots). The two linear regression lines in e) show C change for the different forest floor stock ranges (above and below 25 Mg C ha$^{-1}$). LAU75 is Lauterberg 75 and RAN50 is Rantzau 50.

Overall, there were no significant changes in mineral SOC stock at any depth (Figure 3a), when all sites are pooled together. Unlike coniferous forests, broadleaf forest plots (n = 11) exhibited significant increases in SOC stock in the topsoil (0-5 cm) (3.5 ± 1.9 Mg C ha$^{-1}$, Figure 3b). While it was not significant for SOC stock changes (Table A3), changes in soil C content hinged on the inherent (control) C content (Figure 5). In the mineral soil, the experimental sites that initially had low C contents exhibited increases in C, while sites with already high C contents exhibited decreases. Accordingly, when sites



were classified as having either inherently high C contents (>5.5 % at 0-5 cm, n = 9) or inherently low C contents (<5.5 % at 0-5 cm, n = 17), large differences in SOC stocks between the two categories

became evident in soil profile (Figure 3c). Namely, SOC stocks increased significantly at sites which inherently had low C contents in the control plots (C content <5.5 % at 0-5 cm, Figure 3c). Here, gains

in mineral SOC stocks (0-60 cm) were significantly higher than zero (13.1 ± 4.7 Mg C ha$^{-1}$), although these gains were offset by the SOC losses in the forest floor layers (-10.6 ± 5.6 Mg C ha$^{-1}$). Conversely,

the sites that inherently had high SOC contents in the control plots (C content >5.5 % at 0-5 cm), did not exhibit significant changes in SOC stock at any soil depth throughout the profile, whereby there

was a tendency to have SOC losses throughout the profile (forest floor: -5.6 ± 3.5 Mg C ha$^{-1}$, mineral soil 0-60 cm: -16.4 ± 3.8 Mg C ha$^{-1}$). Next, SOC stocks significantly increased in the 0-5 cm layer in sandy

sites (<50 % sand, Figure 3d), while sites with higher clay and silt fractions exhibited no change in SOC stocks at any depth. In both the forest floor $O_h$ horizon and at 0-5 cm depth, soil C:N ratios decreased

significantly as a result of liming (Figure A6a).

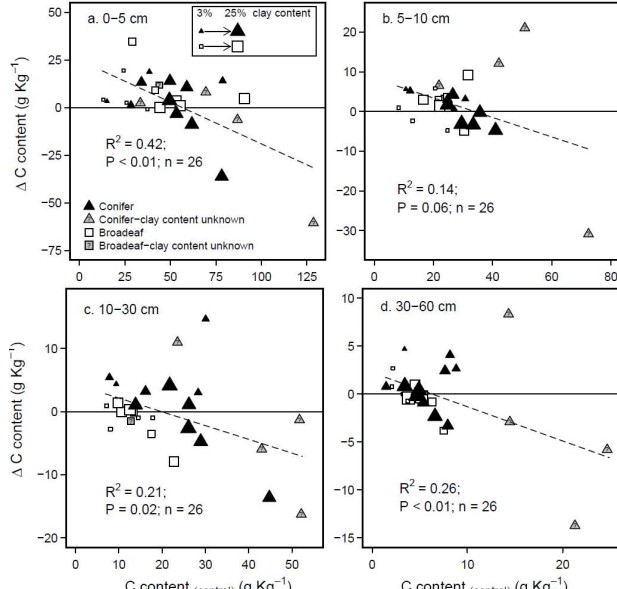

**Figure 5:** Scatterplot diagrams showing SOC contents in the control plots in relation to the liming
induced changes in SOC contents for the different sampling depths in the soil profile. The size of the symbol reflects the amount of clay present at each site. Grey points indicate sites where soil texture
was not known.

### 3.4 Liming effects on soil CO₂ and CH₄ fluxes in beech forests

The soil greenhouse gas flux measurements made prior to re-liming at the DAS 4227 site (indicative of the long-term effects of liming) showed that (1) there were no significant differences in soil respiration



rates between limed and control plots (P = 0.49, Figure 6a), but that (2) methane uptake was twice as

high in the limed plots compared to the control (P < 0.01, Figure 6d-f). Immediately following the re-

liming, soil $CO_2$ fluxes increased and remained consistently higher than the control (P < 0.01, Figure 6a)

for the duration of the measurements. Soil methane uptake on the other hand did not respond to the

liming application, and remained consistently lower than the control (P < 0.01 Figure 6d).

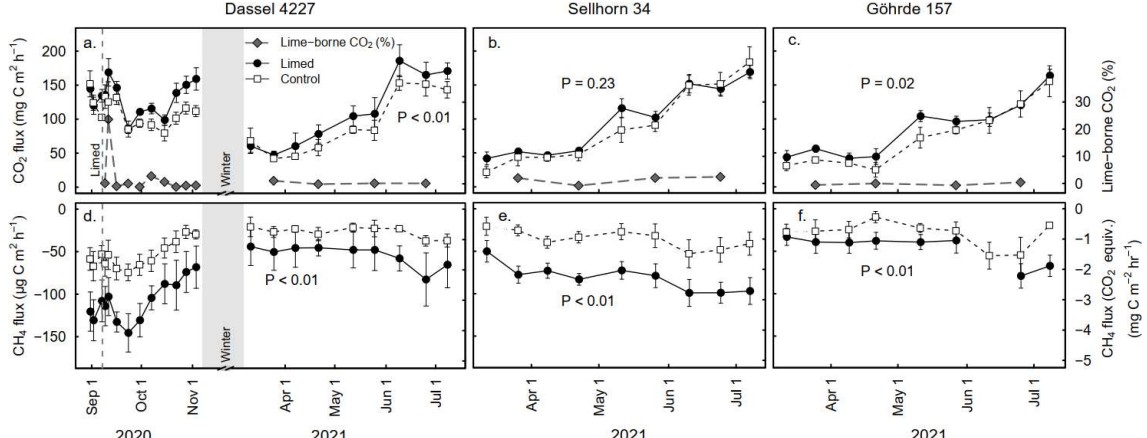

**Figure 6:** Mean (± SE) soil $CO_2$ (a-c) and $CH_4$ fluxes (d-f) at the Dassel 4227, Sellhorn 34 and Göhrde 157
       sites in limed and control plots. The grey line in (a-c) indicate the percentage of lime- derived $CO_2$ of
the total $CO_2$ flux. P-values indicate the significance level between treatments based on LME models.
       At the Dassel 4227 site, the three baseline measurements made prior to re-liming give an indication of
existing long-term differences in soil $CO_2$ and soil $CH_4$ fluxes. At time there were no significant
       treatment differences for $CO_2$ fluxes (P = 0.49) but CH4 fluxes were significant between the treatments
(P < 0.01). At each site 3 Mg $CaCO_3$ ha$^{-1}$ were applied in the late summer of 2020.  Soil $CH_4$ fluxes are
       presented in both actual measured units and $CO_2$ equivalence, based on a global warming potential of
28.

Soil respiration measurements made at the beginning of the growing season of 2021 (6 to 10 months

after liming) at the three sites  (GOH 157, DAS 4227, SEL 34) show that overall soil $CO_2$ fluxes were

significantly higher (23 ± 7 %, P < 0.01) in the limed plots in comparison to the control (Figure 6a-c).

The strength of the liming response however depended on the site, where both GOH 157 and

DAS 4227 exhibited significant increases in $CO_2$ fluxes, while the SEL 34 site did not show any significant

change in $CO_2$ flux (Figure 6b). Overall, soil methane uptake was significantly higher in the limed

treatments (P < 0.01) and was on average two times higher than the control at all three sites (Figure

6d-f).

Using a stable isotope analysis approach, the overall contribution of lime- derived $CO_2$ was low,

averaging 2.7 % of the total $CO_2$ flux in the first two months after lime application at the DAS 4227 site.

At this site, there was only one short-lived lime-derived $CO_2$ pulse immediately after a rewetting event



five days after liming (Figure 6a) which accounted for 23 % of total (biotic and abiotic) $CO_2$ emissions.
The lime- derived $CO_2$ contribution remained negligible the following spring when we measured at the
three sites, averaging $0.7 \pm 0.5$ % (n = 3) of the total $CO_2$ flux.

## 4. Discussion

### 4.1 Liming effects on organic C stocks in the forest floor layers

Over the last three decades, forest floor C stocks have gradually been accumulating in both the limed
and control plots (Figure 2a), with increases most being pronounced at coniferous sites. These gains
likely reflect the influence of elevated N depositions (among other factors) that can (1) enhanced tree
growth accordingly increase litter inputs (Court et al., 2018, Van der Perre et al., 2012) and/or (2)
constrain organic matter decomposition rates (Knorr et al., 2005). The effects of N additions were
362 particularly evident at our coniferous forest plots where sites with higher N deposition had larger
forest floor carbon accumulations (Figure A4).

Considering the biochemical environment plays an intrinsic role in many soil biological processes
(Andersson and Nilsson, 2001; Persson et al., 2021; Melvin et al., 2013), changes in soil pH from liming
can and will cause a cascade of responses that concomitantly affect the net soil C balance. In the
temporal (chronosequence) analysis, the small absolute gains in the forest floor C stocks measured in
the limed plots over time (Figure 2a) were significantly lower than those measured in the unlimed
control plots (Figure 2b), highlighting how lime applications have (in the majority of sites) promoted
organic matter mineralization and offset forest floor OM accumulations. Since overall C stock gains
were comparatively minor (in relation to the control), it indicates that lime applications here helped
maintain stable organic matter decomposition and nutrient cycling rates. These results are further
substantiated in the paired approach analysis, where a larger number of plots were included (Figure
3a). Although this analysis partly contrast the findings reported by the German National Forest Soil
Inventory (which showed that liming decreased forest floor C stocks while unlimed plots remained
unchanged over time (Grüneberg et al., 2019)), both of these studies show the same relative trends:
namely that liming stimulates organic matter mineralization. This too is corroborated by most other
studies (Court et al., 2018; Kreutzer, 1995; Marschner and Wilczynski, 1991; Persson et al., 2021),
whereby some publications (Derome et al., 2000; Melvin et al., 2013) have reported the opposite,
namely that under certain conditions liming can actually increase soil C stocks.

In this study, we found a clear exponential relationship evident between forest floor C stocks and forest
floor layer pH (Figure 1) namely poor sites with acidic pH had high C accumulations in contrast to sites
with higher pH that had lower C stocks. In conjunction with increased microbial-induced SOM



mineralization, it is also likely that increases in earthworm activity, which is known to increase with

liming (Persson et al., 2021), will have assisted the breakup of the litter and the mixing of the organic

matter with soil particles and microorganisms throughout the soil layer (Kreutzer, 1995, Persson et al.,

2021). Next, the improvements in forest floor composition and morphology were also visually evident

at six of the 28 experimental sites, where the humus-form classification improved along the moder to

mull gradient. Moreover, there was also an additive effect of the lime quantity on C stock losses, where

higher lime applications translated to larger C differences with the limed plots (Siepel et al., 2019,

Figure A5). This is not surprising, considering that the more lime applied, the stronger was the

corresponding effect on soil pH (for example, the change in pH in response to the lime's acid

neutralizing capacity at 0-5 cm was highly significant ($R^2$ = 0.43, P <0.01)).

The proportional net change in forest floor C stocks, either C losses or C gains (in relation to the control)

which were observed at a few plots, could best be explained when put in the context of the C stock

present in the control plot. This is because the inherent forest floor C stock (in the control plots) is a

good index of organic matter decomposability (hereafter called the decomposability index) showing

the integral effect of different biochemical drivers (such as pH and litter quality) that regulate SOM

breakdown. For instance, sites with high C stocks had correspondingly acidic pHs (Figure 4a), high C:N

ratios (Figure 4b), and both high C contents (Figure 4c) and high SOM mass (Figure 4d). This contrasts

those sites with inherently low forest floor C stocks which had higher pH, low C:N ratios, low C contents

and thin organic matter layers. When we use the C stock of the control plots as an index of carbon

bioaccumulation, we see that liming effects on forest floor C stocks are most pronounced at sites with

intermediate amounts of carbon (18-35 Mg C ha$^{-1}$), and less prominent at the other ends of the index

(Figure 4e). First, liming additions to sites which had inherently low forest floor C stocks (characterized

by thin SOM layers and high pH) exhibited only small proportional losses in overall C stocks (Figure 4e).

This minor response is because these sites already had relatively high pH values and the addition of

lime did not change the biogeochemical environment dramatically, and accordingly there were no

large changes in forest floor C stocks. Next, further along this decomposability index, sites with

intermediate amounts of carbon exhibited large C losses (up to 75 % decreases). This is because the

application of lime improved the biochemical environment for microbial communities thereby

stimulating organic matter decomposition, which led to strong C losses at these sites. Finally, further

along this decomposability index, at sites having inherently high forest floor C stocks the application

of lime had an increasingly muted effect on C losses, ultimately leading to C gains at some sites (for

example LAU75). Sites at this end of the spectrum were particular poor, having inherently low pH and

thick organic horizons. Here we suspect that more lime had to be applied in order to buffer soil

acidification in an extent that leads to pH improvements favorable for soil microorganisms and other

soil biota. Thus, microbial activity and accordingly also decomposition rates remained more or less



unchanged. We suspect that the inherent biochemical conditions at this end of the spectrum are likely

similar to those reported by Melvin et al. (2013) in hardwood forests in the USA and by Derome et al. (1990, 2000) in spruce and pine stands in Finland, who both report significant gains in SOC stocks as a

result of liming.

### 4.2 Liming effects on organic carbon stocks in the mineral soil

In the mineral soil, liming had a variable response on SOC stocks. While liming may not have induced an overall significant change in SOC stocks at any soil depth (Figure 3b), the direction and magnitude

of net SOC changes in response to liming at each site hinged on the strength of different processes at each site. These are primarily influenced by the sites' biochemical conditions and forest type. The

observed variable response is driven by the dynamic balance in soil carbon accumulation rates, namely organic matter inputs, its stabilization and losses as $CO_2$ or dissolved organic carbon (Jackson et al.,

2017). Considering the broad biophysical spectrum of sites we sampled at, this net C balance (losses versus gains) varied considerably in response to the increases in soil pH and base saturation in the

topsoil. Like in the forest floor layer, SOC losses can be attributed to the stimulation of microbial decomposition of organic matter. The direction and magnitude of the liming-induced SOC stock

changes in the mineral soil (at all soil sampling depths) could however best be explained by the soil's SOC storage capacity and how much carbon was stored therein. Generally, we found that sites with

low inherent soil carbon contents (in the control plots) exhibited SOC increases, while at the other end of the spectrum those sites with inherently high carbon contents, exhibited decreases in SOC (Figure

5). This trend was also observed by van Straaten et al. (2015) after land-use change, and shows that sites with inherently high SOC stocks are more vulnerable to SOC losses than sites which initially had

little to lose. When we separated our dataset into "carbon rich" (SOC content > 5.5 % at 0-5 cm depth) and "carbon poor" sites (SOC content < 5.5 %) we recorded significant increases in SOC stocks in those

sites which initially had low carbon (Figure 3c), but no significant change for sites with initially high SOC stocks. The lack of a significant response in this case likely reflects that we did not sample many sites

with inherently high soil carbon. Considering the "carbon-poor" sites mostly had low clay contents (high sand and low clay), and low soil fertility, the corresponding SOC increases after liming likely

reflect a re-equilibration of the ecosystem carbon cycling dynamics (Figure 3d). We suspect, that C stocks were initially depleted at these sites because sustained acidification over decades which will

likely have constrained aboveground net primary productivity, and accordingly reduced C inputs into the soil. Subsequent improvements in nutrient availability and reduced Al toxicity as a result of liming

likely improved tree growth (Court et al., 2018, Van der Perre et al., 2012), which in turn  increased C inputs into the soil. These suppositions are supported by Grüneberg et al., (2019), who similarly report

that liming led to high C accumulations at sites with low clay contents, and C losses at sites with high clay contents.



Next, improvements in both the biochemical environment and litter palatability will likely have stimulated earthworm bioturbation (Persson et al., 2021), especially considering earthworm abundance is positively related to calcium availability (Hobbie et al., 2006, Reich et al., 2005). And while earthworm activity is known to promote organic matter mineralization (Lubbers et al., 2017), they also foster the stabilization of physico-chemically protected carbon in soil aggregates by building up mineral-protected microbial necromass (Angst et al., 2019). It is also suspected that the decreases in soil bulk densities in the topsoil (Figure A2b) are attributed to this intensified earthworm activity in the liming plots, which will have loosened and aerated the soil improving gas diffusion, therein also incorporating SOM from the $O_h$ into the mineral soil. Although the net effect of earthworm activity on SOC stocks may not be clear (Persson et al., 2021), it may offer an insight into why net SOC stocks significantly increased in the topsoil (0-5 cm) in the broadleaf forest sites (Figure 3a) where leaf Ca increased as a result of liming, but not in the coniferous forests where needle Ca did not improve (data not shown). Another possible mechanism for the measured increases is through Ca-SOM bridging. Here, the divalent $Ca^{2+}$ cations bonded on negatively charged organic matter exchange complexes which stabilize the SOM, thereby reducing the dissolution and mobility of the SOM (Kalbitz et al., 2000) and correspondingly also reducing decomposition processes (Grüneberg et al., 2019; Melvin et al., 2013).

**4.3 Liming effects on soil respiration and soil methane fluxes**

The comparable soil respiration rates measured in the limed and control plots at the DAS 4227 site prior to a third lime application, highlight that (at least at this site) soil organic matter mineralization rates had equilibrated after liming (done 27 years prior, Figure 6a-c). The third application of lime (in August 2020) consequently elicited a pronounced and prolonged increase in soil respiration rates at all three sites (Rosikova et al., 2019). These increases were primarily driven by biotic sources with only a very minor contribution (<3 %) coming from lime- derived $CO_2$ (Figure 6a-c). This is in agreement with Biasi et al. (2008) who measured similarly low abiotic $CO_2$ production in a limed peatland forest in Finland. It is most likely that the resulting improvements in the soil biochemical environment created suitable conditions for microbial populations to mineralize organic complexes, which led to the increased $CO_2$ production. However, like SOC stock responses to lime application, the size (and duration) of the $CO_2$ production increase varied for the three sites. Notably, the two sites with thick SOM horizons (SEL 34 and GOH 157) had smaller and also shorter-lived $CO_2$ flushes than the more fertile site (DAS 4227). This again supports the earlier observations that especially at poorer sites characterized with thick forest floor layers, liming responses may be inhibited by the adsorption of lime to SOM complexes.



Interestingly, long-term soil $CH_4$ uptake in the limed plots was more than twice that of the control plot at the Dassel 4227 site (Figure 6b). Although, we did not take baseline measurements at the other two sites, they too had double the $CH_4$ consumption than their respective control plots after liming. This strong $CH_4$ consumption corresponds to the findings of Borken and Brumme (1997), who attributed this to the fact that liming improves both the soil structure (Bronick and Lal, 2005, Schack-Kirchner and Hildebrand, 1998) and reduces the forest floor layer thickness, which in turn improves $CH_4$ diffusion into the soil. Furthermore, it has been shown that methanotroph abundance and activity is optimal at pHs just below 6 (Amaral et al., 1998). Despite these soils being a relatively large $CH_4$ sink, their $CO_2$ equivalency (global warming potential) nevertheless is still dwarfed by $CO_2$ emissions from organic matter mineralization.

## 5. Conclusions

We hypothesized that liming would lead to decreases in the forest floor layer C stock and that these C losses would be offset, if not exceeded, by significant gains in SOC stocks in the topsoil. Liming indeed resulted in significant decreases in forest floor SOC stocks, but these losses were only partially offset by small gains made in the mineral soil under certain conditions.  However, the question of whether liming enhances forest soil C sequestration is not straight forward. Although there were overall decreases in C stocks in the forest floor, the size of these losses depended on the inherent pH and decomposability of the organic material (before liming). While liming stimulated decomposition at most sites, some poorer quality sites which were characterized by thick organic matter accumulations exhibited either only minor C losses, and in a few plots even C gains. Although there were no significant changes in SOC stocks in the mineral soil as a result of liming, the direction and magnitude of C stock changes here were likewise site-dependent. Specifically, sites with sandy soils and/or inherently low C storage exhibited large increases in SOC stocks as a result of liming, while on the other hand, C rich sites were more predisposed to C losses, suggesting that the SOC stocks here were more vulnerable to decomposition than at sites which had little to lose.

Independent of liming, there is evidence of C accumulation in the forest floor layers over the last few decades (likely a response to elevated N deposition), but liming was able to moderate the amount of C that has become immobilized in the organic matter. Liming-induced increases in mineralization rates seem to last for only a limited amount of time, as seen on the respiration rates of the soil, while the doubling in methane consumption due to liming lasts for several decades. Still, $CO_2$-emissions dwarf the $CH_4$-consumption of the soil.



We can conclude that liming has an influence on forest soil organic carbon stocks. The effect is largest

in the forest floor, where liming counteracts the observed temporal organic matter accumulation (due
to N deposition), thereby reducing nutrient immobilization in the forest floor. In the mineral soil the

effect of liming on soil organic carbon stocks in less pronounced, but there are indications that liming
promotes some carbon accumulation processes in the topsoil. In total, the implications of liming on

forest soil greenhouse gas budgets are small, but highly site-specific.



**Appendix A**

**Supplement**

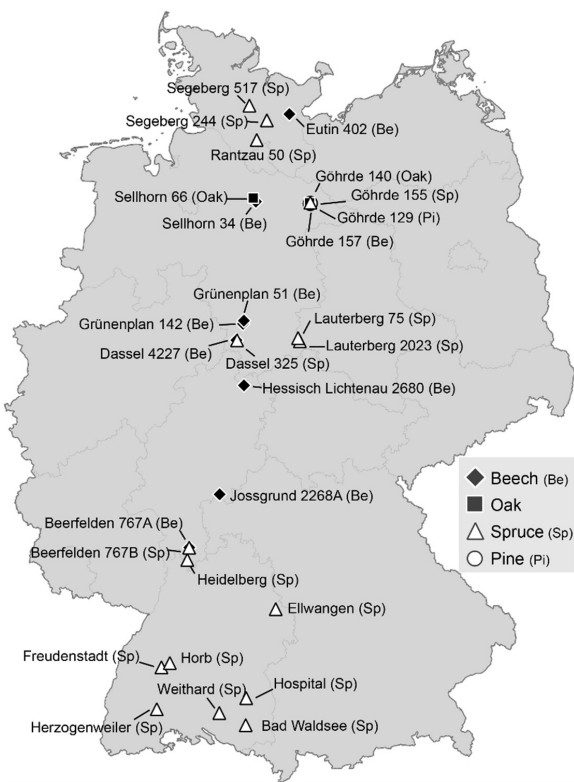

**Figure A1**: Location of the 28 paired liming experiment sites in Germany where soil organic carbon
samples were collected.

**Table A1**: Years when forest floor and mineral soil samples were collected from the different sites

| Site name | Species | Forest floor | Mineral soil |
|---|---|---|---|
| Dassel 325 | Spruce | 1990, 1998, 2004, 2011, 2015 | 1998, 2011, 2015 |
| Dassel 4227 | Beech | 1998, 2004, 2012, 2015, 2018 | 1998, 2012, 2018 |
| Eutin 402 | Spruce | 1990, 1998, 2010 | 1998, 2010, |
| Göhrde 129 | Spruce | 1990, 1998, 2010, 2015 | 1990, 1998, 2010, 2015 |
| Göhrde 140 | Oak | 1990, 1998, 2010, 2015 | 1990, 1998, 2010, 2015 |
| Göhrde 155 | Spruce | 1990, 1998, 2013 | - |
| Göhrde 157 | Beech | 1990, 1998, 2015, 2018 | 1990, 19982018 |
| Grünenplan 142 | Beech | 1990, 1998, 2009, 2015, 2018 | 1998, 2009, 2018 |
| Grünenplan 51 | Beech | 1990, 2019 | - |
| Hess. Lichtenau 2680 | Beech | - | 2012, 2018 |
| Lauterberg 2023 | Spruce | 2000, 2009, 2015 | - |
| Lauterberg 75 | Spruce | - | 1998, 2005, 2015 |
| Rantzau 50 | Spruce | 2000, 2010, 2017 | 2000, 2010, 2017 |
| Segeberg 244 | Spruce | 1990, 1998, 2004, 2017 | - |
| Segeberg 517 | Spruce | 2000, 2010, 2017 | 2000, 2010, 2017 |
| Sellhorn 34 | Beech | 1990, 1998, 2010, 2015, 2018 | 1998, 2010, 2018 |
| Sellhorn 66 | Beech | 1990, 1998, 2010, 2015 | 1998, 2010, 2015 |



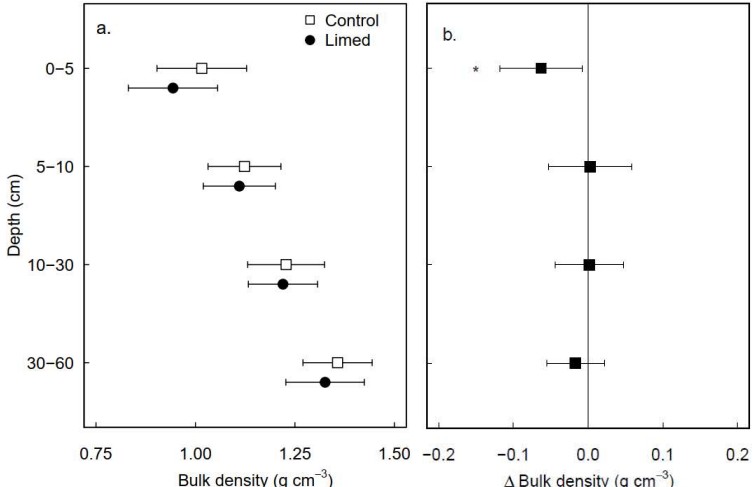

**Figure A2**: (a) Mean soil bulk density in the control plots of conifer and broadleaf forest plots and (b)
differences in soil bulk density between limed and control plots. Error bars indicate the 95 %
confidence intervals based on Student's T distribution. Statistical significance was tested using LME
models for each respective soil depth / layer at P ≤ 0.05 (*) and P ≤ 0.01 (**).

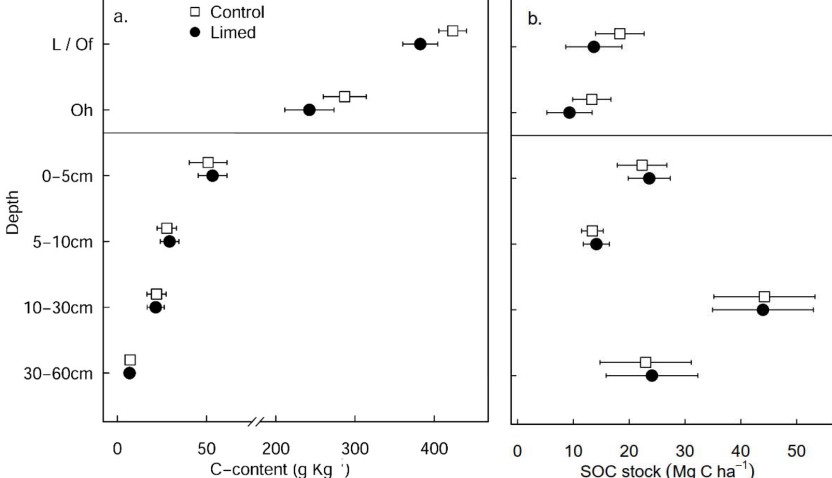

**Figure A3**: (a) Mean SOC contents in the limed and control plots and (b) mean SOC stocks in the limed
and control plots (forest floor layer n=28, mineral soil n=26). Error bars indicate the 95 % confidence
intervals based on Student's T distribution.





**Table A2:** Spearman correlation coefficients of SOC stock in the control plots at different soil depths with explanatory variables (forest floor layer n = 28, mineral soil n= 26).

| | L/O$_f$ | O$_h$ | 0-5 cm | 5-10 cm | 10-30 cm | 30-60 cm |
|---|---|---|---|---|---|---|
| Precipitation (mm a$^{-1}$) | -0.10 | -0.21 | 0.11 | 0.07 | 0.14 | 0.29 |
| Temperature (°C) | -0.26 | -0.06 | -0.16 | -0.24 | -0.37 | -0.31 |
| Elevation (m.asl) | -0.23 | -0.26 | -0.10 | -0.06 | 0.32 | 0.13 |
| N-deposition (kg N ha$^{-1}$) | **0.41\*** | 0.31 | 0.30 | 0.38 | 0.13 | 0.26 |
| Clay (%) | - | - | 0.49 | 0.51 | **0.62\*** | 0.23 |
| Sand (%) | - | - | **-0.62\*** | **-0.56\*** | -0.43 | -0.35 |
| C:N ratio | -0.01 | **0.65\*\*** | -0.05 | 0.11 | 0.13 | 0.32 |
| Base saturation (%) | 0.07 | 0.07 | -0.07 | -0.33 | -0.38 | **-0.48\*** |
| pH (H$_2$O) | **-0.79\*\*** | **-0.67\*\*** | -0.27 | -0.29 | **-0.41\*** | **-0.63\*\*** |

* Indicates a P-value of ≤0.05, and ** indicates a P-value of <0.01

**Table A3:** Spearman correlation coefficients of SOC stock changes (limed minus control) at different soil depths with explanatory variables (forest floor layer n = 28, mineral soil n= 26).

| | L/O$_f$ | O$_h$ | 0-5 cm | 5-10 cm | 10-30 cm | 30-60 cm |
|---|---|---|---|---|---|---|
| **Climate and site characteristics** | | | | | | |
| Precipitation (mm a$^{-1}$) | 0.29 | 0.13 | 0.08 | -0.23 | 0.01 | -0.25 |
| Temperature (°C) | 0.06 | -0.28 | 0.12 | 0.09 | 0.19 | -0.09 |
| Elevation (m. asl) | 0.05 | -0.10 | 0.00 | -0.27 | -0.08 | -0.12 |
| Acid neutralization capacity (kmol$_c$ ha$^{-1}$) | **-0.44 \*** | **-0.46 \*** | -0.08 | **-0.34 §** | 0.08 | 0.01 |
| Nitrogen deposition (kg N ha$^{-1}$) | 0.04 | 0.29 | **-0.34 §** | 0.02 | **-0.40 \*** | -0.17 |
| **Soil properties of the control plot** | | | | | | |
| Clay (%) † | - | - | **-0.53 \*** | 0.00 | -0.31 | -0.16 |
| Sand (%) † | - | - | 0.43 | 0.13 | **0.50 §** | 0.04 |
| SOC stock (Mg C ha$^{-1}$) | **-0.38 \*** | -0.06 | **-0.55 \*\*** | -0.32 | -0.28 | -0.28 |
| C:N ratio | -0.00 | 0.03 | 0.15 | **0.34 §** | 0.14 | 0.02 |
| Base saturation (%) ‡ | 0.11 | 0.04 | -0.27 | -0.08 | 0.00 | 0.15 |
| pH (H$_2$O) | 0.14 | -0.16 | -0.12 | 0.11 | -0.17 | 0.11 |
| **Changes in soil properties as a result of liming** | | | | | | |
| Δ H$^+$ | 0.27 | 0.06 | 0.04 | **0.47 \*** | 0.26 | **0.49 \*** |
| Δ Base saturation ‡ | -0.23 | -0.40 | 0.11 | **-0.33 §** | -0.12 | -0.26 |

Levels of significance: § p < 0.10, * P ≤ 0.05, ** P ≤ 0.01, † n=15, ‡ n=11 in the forest floor layers





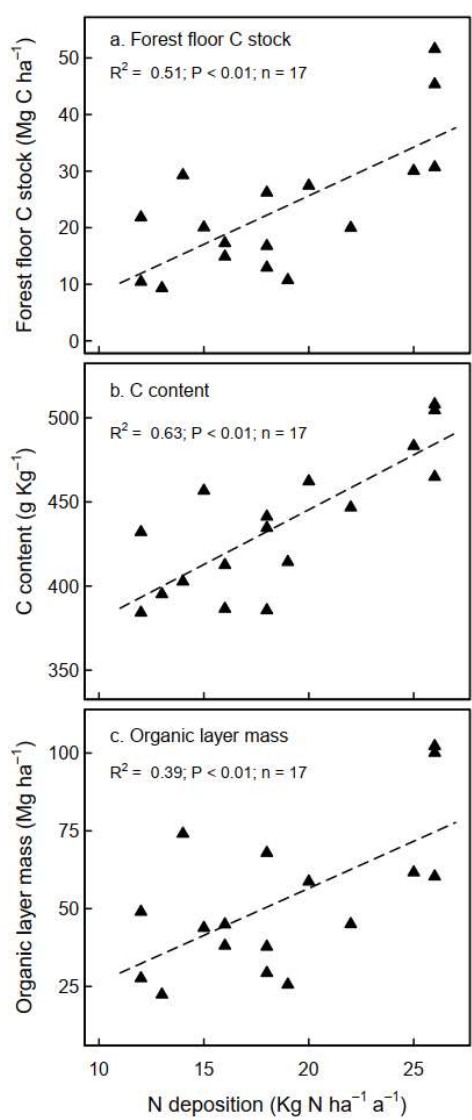

**Figure A4:** Effects of N deposition on (a) SOC stock (b) C content and (c) dry mass of the L/O$_f$ horizon
in unlimed coniferous forests. There was no significant correlation evident with broadleaf forests sites.



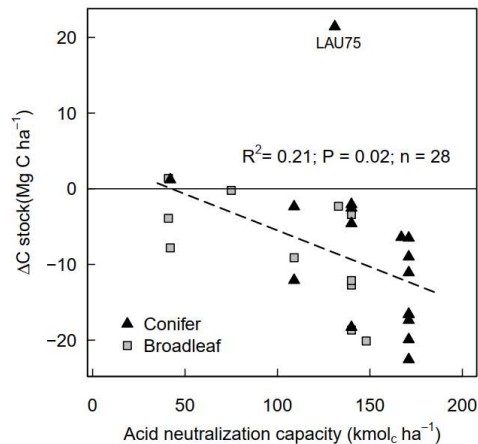

**Figure A5:** Changes in forest floor C stock (L/O$_f$ and O$_h$) between limed and control plots in relation to
liming quantities, expressed as acid neutralization capacity (ANC). LAU 75 is Lauterberg 75.

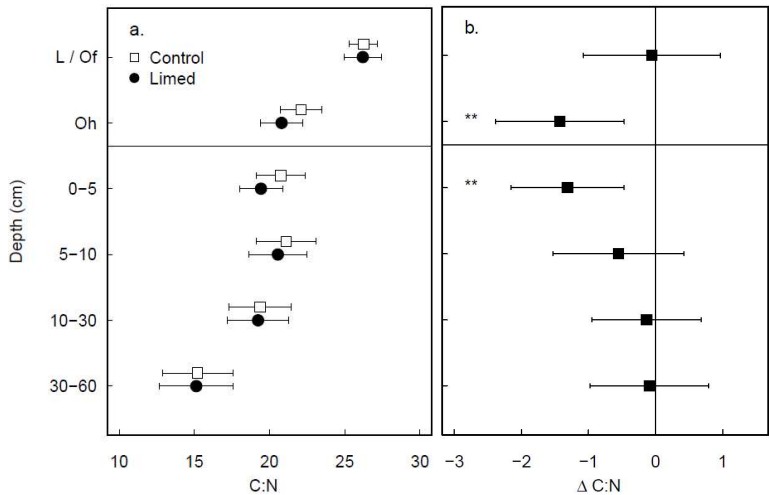

**Figure A6:** (a) Mean C:N ratio of limed and control plots and (b) treatment difference at different
depths in the 60 cm soil profile. Error bars indicate the 95 % confidence intervals based on Student's T
distribution. Forest floor layer n=28, Mineral soil n=26. Statistical significance was tested using LME
models for each respective soil depth / layer at P ≤ 0.05 (*) and P ≤ 0.01 (**).



**Author contributions**

The project was conceptualized UT and DZ. UT, DZ, LK coordinated the data collection activities and
oversaw the maintenance of the liming pairs. GM calculated the lime-derived $CO_2$ emissions. OvS did

the data analysis and prepared the paper. UT, DZ, LK and GM gave critical feedback on the paper.

**Competing interests**

The authors declare that they have no conflict of interest.

**Acknowledgements**

This study was financed by the Agency for Renewable Resources (Fachagentur Nachwachsende
Rohstoffe e.V.) under the grant 28W-B-4-075-01. The authors gratefully acknowledge Dr. Peter

Hartmann, Lelde Jansone and the Forest Research Institute Baden-Wuerttemberg for the soil
biochemical data from eight liming experiment sites. We also kindly acknowledge Dr. Karl Josef Meiwes

and Dr. Jan Evers for valuable input in the data analysis and interpretation. We also thank Lena
Wunderlich for her assistance collecting the soil GHG samples.



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
