# Peer review of "Forest liming in the face of climate change: the implications of restorative liming on soil organic"

_EGUsphere, 2022_

## Author Comment (AC1)

Dear Reviewer #2,

We would like to thank you for your time and for the critical and constructive feedback you have given our manuscript. We have attempted to address your comments in the manuscript as best as possible. References to line numbers refer to the preprint.

The article deals with the effect on liming across different forest types in Germany. Liming was generally performed to reduce the acidity and increase forest productivity. Given the contrasting results present in the scientific literature about this topic, the authors try to clarify the effect of this operation using both space-for-time substitution and chronosequence approaches. The topic is worth to be investigated and suitable for this journal.

The material and method section is very clear and the methods applied both for the sampling (litter layer and mineral soil) are sounds. My only concern is about the sampling of the organic horizon that was collected on a very small area since the root auger used for sampling has a diameter of 8 cm only.

Author's response:

To clarify, at each of the four points where we sampled in each plot we took three individual samples using a root auger (~150 cm$^2$ * 4 = 600 cm$^2$) and pooled them together to make one sample. The picture below demonstrates how and where samples were taken.

To make this clearer we changed the wording in the sentence (lines 115-116) to:

*"At each of the four sampling locations per plot, three samples were taken in close proximity to another for each depth and pooled."*

[Figure]

In the section where the flux measurements are described, some more info are required. For instance, it is no clear to me why the measurements were performed only in the limed sites (prior and after the liming). Why for the baseline it was not used the same approach as for the soil sampling, namely a control site and a limed site? From line 472 it seems that you measured the soil flux in both control and limed plots, isn'it? So, probably some clarification in the text are needed.

Author's response:

To clarify, we measured soil $CO_2$ and $CH_4$ fluxes in both limed and control plots prior and after liming. We have updated the text to make this clearer (line 163).

The Discussion section is well developed and each of the three different subsections clearly address the impact of liming on forest floor and mineral soil carbon stocks. Scientific literature is updated. Figures and tables are easy to understand.

Conclusions are sounds and in line with the hypothesis done at the end of the introduction section.

In general, I found the article very easy to follow and well organized.

Specific comments:

Table 1: Why the soil features in this tables are reported only for the 0-5 cm depth? Are these measures coming from the previous study or are from some historical data? Similarly, why the texture is reported for the 30-60 cm layer?

Author's response:

The soil pH and base saturation values reported in Table 1 were measured from samples taken during the last sampling campaign. We have updated the Table title to reflect this. Next, we reported soil pH and soil base saturation in the upper most soil layer (0-5 cm) because this is where the effect of liming was most pronounced. On the other hand, we reported the soil texture from a deeper soil depth (30-60 cm) because the subsoil texture better reflects the growing medium of both sites. Because of earthworm bioturbation, soil texture differences between limed and control plots were evident in the upper soil layers at some sites.

line 190 – 2044: some more info about the 13 C measurements such as the standards and type of instruments used for measurements would be welcome.

Author's response:

We have elaborated on the section describing the $\delta^{13}C$. The text now reads:

"*The carbon isotope signature ($\delta^{13}C$) of $CO_2$ was determined by isotope ratio mass spectrometry after gas chromatographic separation, the $\delta^{13}C$ of the added lime was analyzed using an isotope ration mass spectrometer coupled to an elemental analyzer, both at the Centre for Stable Isotope Research and Analysis (KOSI) at the University of Göttingen.*"

We did not include information on the standards as the precise $\delta^{13}C$ signature value is not important for the mixing pool model calculation, here only the relative differences between limed and control plots are needed.

Similarly, the samples where these measurements were performed should also be indicated (e.g. sites and number of samples).

Author's response:

We have elaborated the methods section to better explain where and how the samples were taken. The text now reads:

> "The proportion of lime-derived $CO_2$ to the overall $CO_2$ flux, was determined using $\delta^{13}C$ stable isotope approaches and a two-pool mixing model at the same three sites where soil GHG fluxes were measured. Unlike the soil GHG measurements, we collected gas samples for $\delta13CO2$ analysis every second measurement campaign. Samples were collected two minutes and 62 minutes after chamber closure"

Line 254: it would by nice to know how long back in time the historical data are referring. 10 yr? 20 yr? more? From figure 2 it seems that the historical data are referring to the previous 20 yr, isn'it? I appreciate table A1 at the end where all the dates of the previous sampling are indicated

Author's response:

We have updated the text in line 254 indicating when the historical data was collected. The text now reads:

> "At a subset of experimental sites where historical data were available, most dating back to 1990 (Table S1), …"

We have however moved Table A1 to the supplementary section (which is a separate document), in line with the reviewer's earlier comments to streamline the manuscript. While it is 'nice-to-have', this table simply provides background information, and we felt it was not important enough to maintain in the main manuscript.

459-464: the possible impact of earthworms could be partly evaluated looking at the C concentrations. Since earthworms move vertically more homogeneous C distribution between the different depths should be present is sites where earthworms' activity is higher compared to control plots where liming was not applied.

Author's response:

We looked into this and found that indeed there were slightly (significantly) higher C contents in the limed plots at 0-5 cm in deciduous forest sites (8.2% ± 7.1 %; mean ± 95% confidence interval). However no trend was evident when all plots were included. Below 5cm depth no difference was evident.

We have now mentioned this in the discussion section (underlined text).

> "Next, improvements in both the biochemical environment and litter palatability will likely have stimulated earthworm bioturbation (Persson et al., 2021), as is evident from the higher C-contents measured in the top 5 cm of soil in the limed plots in the broadleaf forest plots (data not shown)."

---

## Author Comment (AC2)

Dear Reviewer #1,

We would like to thank you for your time and for your critical and constructive feedback. We have attempted to address your comments below and have integrated many of your comments in the manuscript. References to line numbers refer to the preprint.

Dear Editor,

These are my comments about the paper 'Forest liming in the face of climate change: the implications of restorative liming on soil organic carbon in mature German forests', of Oliver van Straaten et al., submited to EGUsphere.

It is a good paper, worth to be published. However, I have four main comments, which perhaps need a bit of attention from the authors, and also several minor complaints, mostly about small details of writing, which can be solved by authors (I assume) without special problems.

MAIN COMMENTS

Authors collect a nice set of paired plots, covering a variety of climates and situations, and thus they may explore their dataset in search for a variety of correlations and comparisons. Perhaps too much; this is evidenced by the huge amount of figures and tables, many of them are placed in an 'Annex' chapter: by the way, to me it is not clear whether this 'ANNEX' equals to a 'Supplementary Material': will these figures appear in the main core of the published paper?

Author's response:

We acknowledge that there are a large number of figures and tables in the manuscript, especially when combined with the appendices. Specifically, the preprint has six figures and one table in the main body of the manuscript, while the appendices has six figures and three tables.

While these graphs and tables are full of information and are perhaps "nice-to-have", we agree that having too many figures can be overwhelming for the reader. According, after serious consideration and prioritization of the results, we reorganized how (and where) we present the figures and tables. Specifically, we have moved one figure (Fig. 5) out of the main body of the text and placed it in the appendices section. This figure has additionally been simplified, by removing the texture effects (where point sizes were proportional to clay percent). Next, we have removed two figures and three tables from the appendices section and moved them to the supplementary section (which is a separate document). By moving these figures and tables, the aim was to streamline the results and declutter the manuscript, while at the same time not losing important background information for the avid reader.

Specifically, we:

1. We moved the following figure from main body and "downgraded" it to appendices:
- Figure 5: Scatterplot diagrams showing SOC contents in the control plots in relation to the liming- induced changes in SOC contents for the different sampling depths in the soil profile.

2. Next, we moved the following from appendices to supplementary (stand-alone) section :
- Figure A3: (a) Mean SOC contents in the limed and control plots and (b) mean SOC stocks in the limed and control plots

- Figure A6: (a) Mean C:N ratio of limed and control plots and (b) treatment difference at different depths in the 60 cm soil profile.
- Table A1: Years when forest floor and mineral soil samples were collected from the different sites
- Table A1: Spearman correlation coefficients of SOC stock in the control plots at different soil depths with explanatory variables
- Table A2: Spearman correlation coefficients of SOC stock changes (limed minus control) at different soil depths with explanatory variables

- Additionally, we have removed Figure A2a completely. This showed the differences in mean soil bulk densities of the conifer and broadleaf plots.

In the updated version, we nevertheless still have five figures in the appendix, as these provide valuable background information that improves the transparency of our results. The figures in the Appendix are:

- Figure A1: Map of the 28 experiment sites in Germany
- Figure A2: differences in soil bulk density between limed and control plots
- Figure A3: Effects of N deposition on (a) SOC stock (b) C content and (c) dry mass of the $L/O_f$ horizon in unlimed coniferous forests
- Figure A4: Changes in forest floor C stock ($L/O_f$ and $O_h$) between limed and control plots in relation to liming quantities
- Figure A5: Scatterplot diagrams showing SOC contents in the control plots in relation to the liming- induced changes in SOC contents for the different sampling depths in the soil profile

To answer the reviewer's question: the appendices are appended to the bottom of the manuscript, while supplementary information is a separate document.

The huge number of relationships studied makes difficult for authors maintain the focus. A reduction in the number of topics treated could help to make the paper easier to read. See below, for instance, my suggestion about reducing the space devoted to soil textural effects.

Author's response:

In this paper, we have comprehensively examined the effects of liming on soil C dynamics. This is naturally quite complex since the application of lime drives a cascade of chemical, physical and biological responses, each with implications on soil carbon. Moreover, the analysis is complicated by the fact that many variables have counteracting effects: while one variable may promote soil C build up, another variable will enhance decomposition of organic matter. The resulting paper accordingly presents a lot of statistical analyses, graphs and interpretation.

Nevertheless the reviewer is correct in his/her criticism and we have taken measures to further improve the focus of the paper. We have accordingly made the following changes:

- Streamlined the manuscript to minimize speculation of different mechanisms
- As mentioned above in greater detail, we have moved tables and figures out of the main part of the manuscript (into either the appendices or supplementary) to declutter the main message. In total, we moved one figure out of the main text and into the appendices, next we reduced the number of figures and tables in the appendices from nine to a total of five.
- We simplified Figure 5 to exclude the soil texture aspects (as suggested by the reviewer)
- Removed sentence on additive effect liming has on pH

Four main aspects should be clarified:

a) How relevant are the changes in SOC stocks?

An important detail (very relevant to me, at least) remains unclear, after reading the paper. Liming involves applying inorganic carbon to the soil: being the soils quite acidic, the added carbonates are lost to atmosphere as CO2. Therefore, liming involves making soils a source of C, not a sink. At least for a time. The long-term increases in SOC stocks in limed plots (relative to control plots), do they compensate the initial C losses? In other words: the inorganic C added when liming, lost in the following years, is lower or higher than the extra SOC sequestered thereafter?

Author's response:

If we understand correctly, the reviewer would like to compare the magnitude of $CO_2$ released from lime dissolution with the quantity of $CO_2$ released or sequestered as a result of liming. This is quite difficult to answer, as soil C stocks only represent a part of the entire ecosystem C balance, and a complete C budget of the ecosystem would be necessary to adequately answer this To do this we would need to include measurements of DOC fluxes, above- and belowground biomass production (and other measurements, i.e. measure $CH_4$ uptake over longer time frames). Now even if we had all this data, the error uncertainties of the data would overshadow the $CO_2$ emissions from the lime. Next, the amount of C added to the ecosystem as lime represents a very small proportion of ecosystem net primary productivity, especially if one considers a decadal timeframe. Let me give an example to compare it with soil $CO_2$ respiration:

- A site is limed with 3 t/ha of $CaCO_3$. This equates to 1.3 t C / ha. (this is standard in German forests)
- Now compare this with an average annual soil respiration rate typical for a German forest: 8.7 t C/ ha
- Assuming all the lime is dissolved in the first year, lime would have contributed 13% (= 1.3/(8.7+1.3)) to the annual flux in the first year. However when compared to soil $CO_2$ respiration over one decade, the lime-derived contribution becomes negligible to less than 1.5% (and 0.7% in two decades).

On the other hand, lime-derived $CO_2$ becomes relevant for the second part of the study (for the short-term $CO_2$ fluxes), and this is why we used stable isotope approaches to quantify this.

b) Inorganic carbon was not analyzed

This is a detail that surprises me, to be honest. Why authors did not analyze carbonates in their samples; at least, in the plots that received lime in the past. Part of the carbon in these soils may be carbonate remaining from the added lime. Certainly, after decades, one could expect that all added carbonate has become CO2 and released to the atmosphere; but figure 6 (and also lines 350-353) clearly show that the loss of added lime is extremely slow. On the other hand, as shown in Table 1, some sites were limes up to 3 times. The persistence of lime residues in some places (particularly when soil pH reached almost neutrality) is envisageable. I ask authors to make a rapid screening of some samples (in limed plots) to ensure the absence of inorganic carbon in their samples. Just to be sure that the slight increase in SOC stocks in limed plots (in mineral soil horizons) is due to organic carbon.

Author's response:

We did not ignore carbonates as these were measured when the soil pH ($H_2O$) was higher than 6.2. This however consisted of <2% of the entire dataset. Of these 21 samples, carbonate contents were very small (~3% of the total C stock).

To demonstrate that carbonates are essentially absent at lower soil pH we have taken another dataset (second German National Forest Soil Inventory for Lower Saxony, n=100 sites) and plotted the calcium carbonate concentrations against soil pH ($H_2O$). In this graph it is clearly evident that there are no carbonates present in soils with a pH below 6.5.

[Figure]

Next, it is recognized that the dissolution of lime occurs relatively quickly in acidic soils. For example, Kreutzner (1995) found that lime dissolution follows an exponential curve with time. Based on the exponential function reported in the Kreutzner (1995) article (Mass (Tons/ha) = $3865 * e^{(-0.643 * time)}$), 51% of the lime will be gone within one year after liming, 86% after 3 years, and 99.8% within 10 years. See figure below which is based on Kreutzner (1995).

[Figure]

The actual lime dissolution rate of course depends on different factors including soil pH, soil moisture, temperature, acid deposition, thickness of the forest floor layer, the granule size of the liming material, and the proportion of calcite to dolomite.

Considering that at most sites we collected the soil samples 20 to 30 years after the last liming event we can be quite certain that there are no remaining carbonates left in the soil. At six sites, soil samples were collected three to five years after the last liming event. Even here however, we expect that lime carbonates will be largely gone. If we assume that lime at our sites have similar dissolution rates as Kreutzner (1995) there would be between 4 and 14% of the original carbonates left over at the site (see graph above). Considering we applied 3 t/ha of lime during these applications, there would be between 0.05 and 0.19 Mg C ha$^{-1}$ of the carbonates still in the soil. This is a tiny fraction compared to the C stored in the forest floor layer (~30.7 Mg C ha$^{-1}$).

Even if the dissolution rates were far slower than those measured by Kreutzner (1995), the lime derived carbon would still be a fraction of the overall C-stocks. Let me demonstrate this with an example. Let's assume that 25% of the carbonates still remained in the soil after a liming event (this is quite unrealistic considering the soils at these sites were all very acidic). If the site is limed with 3 tons / ha of $CaCO_3$ (which is standard in Germany) there would 0.33 Mg C $ha^{-1}$ of lime derived C left in the soil. In comparison to the C stocks of the forest floor (~30.7 Mg C $ha^{-1}$) this is still very small, and within the error margins of the soil C measurements themselves.

We have now added a sentence into the methods section explaining why we did not measure inorganic carbon.

> *"Carbonates were measured in soil samples that had a pH ($H_2O$) greater than 6.2. This however consisted of just 21 samples (<2% of the complete dataset), and carbonate contents were a fraction of total soil carbon."*

c) Using litter SOC stock as SOC stability indicator

All plots are placed in humid-temperate forest ecosystems: a large variety in climatic constraints can be discarded, therefore (If I am wrong, then consider including climatic constraints in your analysis). In these conditions, the idea of taking litter SOC (including Oh horizon) as indicator of OC decomposability in these horizons is right. The higher the decomposability, the lower SOC stock in these horizons; but with an additional condition: provided that litterfall is the same in all cases. Or, at least, very similar. This second part of the idea is problematic, for litterfall was apparently not measured in the plots (it does not appear in Table 1). Thus, SOC stock in organic horizons, alone, is a too rough approach to the decomposability of the incoming litter. Note also its problem as indicator: the higher the stock, the lower its decomposability, i.e., the sign is opposite. SOC stock in litter should be an indicator of stability, rather than decomposability.

I have no problem with the use of total SOC stock in litter for studying its relationships with SOC increase, liming, etc. Nice relationships are obtained (Fig. 4). But without identifying it as 'decomposability': in the absence of data about litterfall, such an identification is risky, in my view. In figure 4, for instance, I suggest removing the words 'Organic matter decomposability index' from the 'X' axis. Use simply 'Forest floor C stock in the control plot (Mg C ha-1)', perfectly exact and more adequate.

Apply this to the rest of the text: I suggest authors avoiding (or at least refining) the 'decomposability' in the discussion about the effect of liming in SOC stocks.

Author's response:

The reviewer raises a good point. We have accordingly removed mention of the "decomposability index" and used the terminology suggested by the reviewer namely that the SOC stock in litter is an indicator of "stability". We have for instance in the caption in Figure 4 written:

> *"These four parameters show how the forest floor C stock in the control plots are a good measure of organic matter stability."*

In the discussion we wrote:

> *"This is because the inherent forest floor C stock (in the control plots) is a good measure of organic matter stability showing the integral effect of different biochemical drivers (such as pH and litter quality) that regulate SOM breakdown."*

d) Textural effects

Figure 5 is nice in order to suggest a role for soil texture in the changes of SOC stock related to lime addition. But the conclusions are unclear to me. If the total number of points is n = 26, given the natural variability, perhaps this is unavoidable. Note also that fine soil textures (particularly: high clay content) result often in higher SOC stocks. In figure 5 all points are considered altogether for the regression; Figure 3d is perhaps better in order to suggest a role for soil texture.

Author's response:

We agree with the reviewer that including texture in Figure 5 complicated the story. We have accordingly simplified the figure to exclude clay percent. (For reference: in the preprint, the sizes of the dots were proportional to clay percentages).

Next, we have also downgraded this figure to the Appendix section. The updated figure now looks like this:

[Figure]

Lastly, as the reviewer suggests, the discussion section now only emphasizes the importance of soil texture in relation to Figure 3d (the depth profile graph).

May I suggest you perform a figure similar to 3d, but with clay contents (say, < 14% and >= 14%?). Note that the high number of sites for which soil texture was not available (11 out of 26) suggest being prudent about spending a big effort on this point.

As requested, we made a similar graph for clay contents. See the comparison below between the two graphs below: left graph showing the depth SOC stock depth profile separated in two different clay contents versus in the right graph where it is separated into two different sand contents.

Given that there is a much larger spectrum of sand textures (2-88%) than clay (2-19%), and that the 50% sand mark separated the sample size roughly in half (n=7 vs. n=8), I think sand percent is a better texture metric to use. Because of the broader sand content spectrum the results produced were also clearer.

[Figure]

d.

Clay % (0–5 cm)
■ < 14 % (n=9)
▲ > 14 % (n=6)

d.

Sand % (0–5 cm)
■ < 50 % (n=7)
▲ > 50 % (n=8)

L / Of

Oh

0–5

5–10

10–30

30–60

−33% *
−37% *

−60% *
−76% **

* 38%

Δ SOC stock (Mg C ha$^{-1}$)

Δ SOC stock (Mg C ha$^{-1}$)

SPECIFIC COMMENTS

Line 21 (in abstract). 'Liming however largely offsets this organic layer buildup'. What does it mean, exactly? That the amount of carbon accumulated in forest floor is lower than the C released (as CO2?) upon liming? Or precisely the contrary? Do you refer to losses of organic C or of inorganic C (carbonates)?

Author's response:

I have attempted to improve the clarity of the text. It now reads:

> "Overall, we found that forest floor C stocks have been accumulating over time, particularly in the control plots. Liming however largely offsets organic layer buildup in the L/Of layer, and forest floor C stocks remained unchanged over time in the limed plots. This, in turn meant that nutrients remained mobile and were not bound in soil organic matter complexes."

I am referring to organic C.

Line 21. '...which means that nutrients remain mobile and are not bound in soil organic matter complexes'. I do not see how this statement arises from the previous one.

[Actually, note that if you remove lines 20-22, and start directly with 'Results from the paired plot analysis showed...', the paragraph runs perfectly.

Author's response:

The two sentences in lines 20-22 describe the temporal dynamics over time in the limed and control plots and how they compared. This is a central finding and cannot be removed (as reviewer suggested). The findings described tells us that the application of lime helped maintain nutrient cycling rates, instead of being immobilized within the forest floor which is what will have happened in the unlimed control plots. As background, a major concern raised by liming critics is that lime application results in a rapid mineralization of the forest floor layer, which in turn causes a net nutrient loss from the ecosystem, leading to nutrient deficiencies. This result essentially quashes that concern.

Line 37. Even though?

Author's response:

Corrected, as suggested

Line 53. '...stand age), (3) the application of lime...'. Better?

Author's response:

Corrected, as suggested

Line 54. '...and (4) the ongoing acidification from...'. Better, again?

Author's response:

Corrected, as suggested

Line 58. 'While it is broadly reported that...'

Author's response:

Corrected, as suggested

Line 65. 'Liming-induced changes in nutrient stoichiometry...'

Author's response:

Corrected, as suggested

Line 95. '...meaning that these sites...'

Author's response:

Corrected, as suggested

Line 115. I understand, therefore, that you obtained at each plot four (4) composite samples per depth, each of these obtained by pooling three (3) samples. Ok?

Author's response:

Yes that is correct. To make this clearer we changed the wording in the sentence (lines 115-116) to:

> "At each of the four sampling locations per plot, three samples were taken in close proximity to another for each depth and pooled."

Lines 117-119. I am surprised by the fact that authors did not analyze carbonates in the limed samples. Apparently they are confident that all added lime has disappeared. To me, a verification of this point woul have been welcome. Otherwise, the risk of overestimations of organic C in the limed plots cannot be discarded.

Author's response:

Here we added a sentence to explain that we did measure carbonates, when soil pH was higher than 6.d.

> "Carbonates were measured in soil samples that had a pH ($H_2O$) greater than 6.2. This however consisted of just 21 samples (<2% of the complete dataset), and carbonate contents were a fraction of total soil carbon."

Line 162. '...three beech forest sites: Dassel 4227 (DAS 4227)...' (etc).

Author's response:

We have corrected the sentence.

Line 245 (caption of figure 245). This is (to me) a very relevant detail: in some cases, liming causes the disparition of the Oh horizon. Can you write a couple of lines about this detail? In how many cases does this happen? Could be another explanation, besides the pH increase?

Author's response:

We have added a sentence to the discussion of changes in humus types as a result of liming. We wrote:

> *"Similarly, at seven sites, the application of lime meant that the humic horizons ($O_h$) either did not develop or perhaps were lost over time, indicative of comparatively faster organic matter mineralization rates and / or earthworm bioturbation.*

Line 253. Why is this not shown? This is extremely interesting!

Author's response:

The reviewer is referring to the sentence: *"In the subsoil (30-60 cm), SOC stocks exhibited a similar exponential decay relationship with soil pH as the forest floor layers (data not shown)."*

Although I agree with the reviewer that this is an interesting result, (it highlights the importance of soil pH in regulating soil C stocks in the mineral soil), I would however argue that this result is comparatively minor for two reasons:

(1)  The subsoil was overall less affected by liming, and therefore including it would deviate from the main objectives of the paper. As the reviewer mentioned, the paper already includes a lot of information and it is simply not possible to include everything.
(2)  While the relationship is statistically significant, there is still some noise in the relationship (see graph below).

[Figure]

Lines 259-261. I accept that the difference between control and limed plots did not reach significance, at any depth. Nevertheless figure 2 shows that, even though the variability was very high, SOC sequestration values were on average consistently higher in limed plots. Could this be mentioned somehow?

Author's response:

Figure 2b actually shows that there was a significant difference between limed and control plots. This is indicated with a star. The reviewer is however correct in that this result was not discussed in the discussion section. We have added the following sentence:

*"In the temporal analysis (Figure 2b) we measured increases in SOC in the topsoil (5-10cm) over time similar to, but less pronounced than those reported by the German Forest Soil survey (Grüneberg et al., 2019)."*

Line 307. Perhaps 'partly compensated' rather than 'offset'. If the SOC losses in the forest floor were lower than the gains in mineral soil, then the net balance is still positive, not? If you want, you can write 'partly offset'.

Author's response:

Corrected, as suggested.

Line 364. 'Because the biochemical environmental plays...' (better?)

Author's response:

I personally believe that the sentence is grammatically incorrect if it starts with "Because". I have left it as it is

*"Considering the biochemical environment plays an intrinsic role in many soil biological processes (Andersson and Nilsson, 2001; Persson et al., 2021; Melvin et al., 2013), changes in soil pH from liming can and will cause a cascade of responses that concomitantly affect the net soil C balance."*

Lines 466-470. The importance of calcium for stabilizing soil organic matter, widely known in South Europe (where calcareous environments are common, and calcium-rich soils, too), has been largely underestimated in studies about Central Europe. I acknowledge the mention of Ca2+ as a stabilizer of organic matter in these lines; but I rather regret such a small space given by authors to this explanation, which is to me the most obvious one to account for the increases in SOC stocks after liming. Carbonates are a huge source of calcium for soils, and Calcium (rather than carbonates themselves) the probable main reason for the accumulation of SOC in carbonate-rich soils.

Author's response:

Unfortunately, we have no measurement data to evaluate the importance of Ca-SOM bridging in this experiment, and accordingly we could only speculate that this "may" be a stabilizing mechanism. Therefore, in the interest of streamlining the manuscript (mentioned above by the reviewer) and not getting caught up in further speculations that could dilute the message of actual verified facts, we have opted not to elaborate further on this point.

Line 477. The very minor contribution of lime-derived CO2 to the total CO2 efflux should be taken as a sign of the relative resistance of lime to disappear in a context of acidic soils. The absence of inorganic carbon in soil samples should be verified. Otherwise, the data about SOC stocks may have been overestimated, at least in the limed plots.

Author's response:

I would argue that our isotope measurements actually show the opposite: namely a rapid dissolution of the $CaCO_3$ when it came in contact with water. This was evident in our measurements taken on the day immediately after a raining event. Unfortunately, we did not measure more intensely in the hours immediately after the rain event, to calculate the amount of $CO_2$ lost during that rewetting event.

Lines 484-486. If such an adsorption of lime to SOM complexes did effectively occur, then my complaints about the need of analyzing inorganic carbon in the soil samples makes even more sense.

Author's response:

We have amended this sentence, because we recognize that it was not completely correct. We suspect that C stabilization through $Ca^{2+}$ - SOM bridging was not likely the mechanism responsible for the lack of response at the two sites, but rather relates to other factors affecting organic matter decomposition rates. As Figure 4e shows, sites with thick organic matter layers (in the control) show less pronounced response to liming, because more lime has to be applied to buffer soil acidification to significantly improve soil pH favorable for soil microorganisms and other soil biota.

We have adjusted the sentence to:

> *"This again supports the earlier observations that especially at poorer sites characterized with thick forest floor layers, liming only moderately improves organic matter mineralization rates."*

FIGURES

Must be improved, definitively. In my computer I put the image at 150 % size, and even at such scale I had difficulties in seeing some figures.

Author's response:

We downgraded the image quality of the figures in the preprint so the overall file size would not be too large. For the final version of the manuscript will be upload high resolution (600 dpi) images.

Figure 3 will become almost illegible, for instance. Putting the four sub-figures in line makes all four very small, and letters are almost illegible. I suggest you reconsider the design: instead of four figures in a single line, make a set of four figures, placed in square (two in the upper line, two in the lower line).

Author's response:

We have adjusted the configuration of Figure 3 as suggested by the reviewer.

Figure 4 will suffer the same problem. The four small figures at the top are barely legible in its current form, and in the printed version of the paper cannot be read, quite simply (I verified it). Again, reconsider the design. Also, note that the Figure 4e is different from the others (4a to 4d), and could well be a separate figure.

Author's response:

We have reformatted this figure so that the top graphs (a-d) are larger and more legible. Considering we already have six figures in the main text, we felt that separating this figure into two would be too much. Furthermore, I am of the opinion that the top part of this graph provides valuable information relevant for the interpretation of the bottom graph (e). Essentially, the top graphs makes it clear that the forest floor C stock is an appropriate measure to describe site quality, which in turn is central to understanding why there are differences in liming induced changes in forest floor C stocks at different sites.

[Figure]

Richer sites      vs.      Poorer sites

pH (H₂O)

a.    $R^2 = 0.56; P < 0.01; n = 28$)

C:N

b.    $R^2 = 0.30; P < 0.01; n = 28$)

C content (g Kg⁻¹)

c.    $R^2 = 0.43; P < 0.01; n = 28$)

OL mass (Mg ha⁻¹)

d.    $R^2 = 0.60; P < 0.01; n = 28$)

Percent change in forest floor C stock (%)

e.

$R^2 = 0.49$
$P = 0.01; n = 11$

▲ Conifer
☐ Broadleaf

Forest floor C stock in the control plot (Mg C ha⁻¹)

Graphs in a-d show the parameters (pH, C:N ratio, C content and OL thickness) that are responsible for the amount of C stored in the forest floor (on the x-axis).

Interpreting the meaning of the lower part of the graph is supported by upper part of the figure (a-d).

In combination with the upper parts of the graph, one can see that the liming response depends on the site quality (a product of different variables)